# Integrated Transcriptome and Metabolome Dynamic Analysis of Galls Induced by the Gall Mite *Aceria pallida* on *Lycium barbarum* Reveals the Molecular Mechanism Underlying Gall Formation and Development

**DOI:** 10.3390/ijms24129839

**Published:** 2023-06-07

**Authors:** Mengke Yang, Huanle Li, Haili Qiao, Kun Guo, Rong Xu, Hongshuang Wei, Jianhe Wei, Sai Liu, Changqing Xu

**Affiliations:** Institute of Medicinal Plant Development, Chinese Academy of Medical Sciences and Peking Union Medical College, Beijing 100193, China

**Keywords:** plant gall, gall mite, mite–plant interaction, gall development, molecular mechanism, transcriptome, metabolome, phytohormones

## Abstract

Galls have become the best model for exploring plant–gall inducer relationships, with most studies focusing on gall-inducing insects but few on gall mites. The gall mite *Aceria pallida* is a major pest of wolfberry, usually inducing galls on its leaves. For a better understanding of gall mite growth and development, the dynamics of the morphological and molecular characteristics and phytohormones of galls induced by *A. pallida* were studied by histological observation, transcriptomics and metabolomics. The galls developed from cell elongation of the epidermis and cell hyperplasia of mesophylls. The galls grew quickly, within 9 days, and the mite population increased rapidly within 18 days. The genes involved in chlorophyll biosynthesis, photosynthesis and phytohormone synthesis were significantly downregulated in galled tissues, but the genes associated with mitochondrial energy metabolism, transmembrane transport, carbohydrates and amino acid synthesis were distinctly upregulated. The levels of carbohydrates, amino acids and their derivatives, and indole-3-acetic acid (IAA) and cytokinins (CKs), were markedly enhanced in galled tissues. Interestingly, much higher contents of IAA and CKs were detected in gall mites than in plant tissues. These results suggest that galls act as nutrient sinks and favor increased accumulation of nutrients for mites, and that gall mites may contribute IAA and CKs during gall formation.

## 1. Introduction

Plant galls are among the most emblematic examples of the manipulation and reprogramming of plant development produced by the gall-inducers [1]. There are many kinds of gall inducers in nature, such as bacteria, fungi, nematodes, mites and insects, but the most common gall inducers are insects, followed by mites [2]. The induction of galls is based on DNA transfer in some species of galling bacteria; however, the precise process underlying gall formation by arthropods is still unclear [3]. Research on the mechanism of gall induction in the case of arthropods has always focused on the gall inducers and suggests that gall formation is triggered by the action of chemical substances secreted by them, such as phytohormones [4] and other compounds [5,6,7]. However, the mode of action of these chemical compounds, and the general mechanism by which the gall inducer manipulates plant growth and physiology, are still largely unknown [8]. Furthermore, the chemical hypothesis is highly unlikely to provide a comprehensive explanation of the mechanisms of induction and morphogenesis of these structures due to the intricacy of the induction process and growth of plant galls [4]. The interaction between a plant and an herbivore involves complex networks of molecular and physiological processes within and between the organisms, so understanding plant–herbivore interactions at a system level is indispensable for elucidating the mechanisms underlying the interactions [9]. Therefore, the responses of plant genes and metabolites during the induction process and development of galls are more important for revealing the induction mechanism of galls caused by arthropods.

Due to the unique shapes and wide range of variation of galls, they have been the subject of much attention and research [10]. Gall inducers appear to hijack the plant development system to generate a novel structure in plants [1]. Phytohormones related to cell growth and cell division, such as auxins (AUXs) and cytokinins (CKs), generally increase in galls [11,12]. Tissues inside galls are usually rich in sugar, proteins, amino acids and other nutrients [13,14,15], and tissues outside galls are abundant in phenols, flavonoids and other secondary metabolites [16,17,18,19]. Meanwhile, the activity of oxidative enzymes such as superoxide dismutase (SOD), catalase (CAT), peroxidase (POD) and polyphenol oxidase (PPO) [20,21], and defense-related phytohormones such as salicylic acid (SA) and jasmonic acid (JA) [22], are also enhanced during gall development. Due to this property of galls being enriched in certain metabolic substances, they have long been widely used in industry and medicine [23]. Moreover, these dramatic modifications of host plants caused by gall inducers are usually accompanied by extensive reprogramming of plant gene expression [24]. Specifically, the expression modification of these genes affects the plant’s defense responses [25,26], hormone pathways [27,28], cell cycle, cell wall and cytoskeleton organization [29,30], developmental processes [31,32] and metabolic reprogramming [19,33]. Transcriptomic and metabolomic approaches can provide deep insights into the complexity of gall inducer–host plant interactions and offer reliable, multidimensional molecular biology information for studying and controlling gall inducers [17,27,34]. However, most research has focused on galls induced by insects and nematodes [24], and there has been relatively little research on gall-inducing mites [35]. The mechanisms and specific genes involved in the mite gall-inducing process may differ from those involved in insect gall induction, while they may provide valuable insights into the mechanisms of gall formation and the interactions between gall-inducing organisms and their host plants.

Gall-inducing mites belonging to Eriophyoidea are called gall mites; they are the smallest terrestrial phytophagous arthropods, with body lengths usually varying from 100 to 200 μm, and appear worm-like [36]. As a major group of gall inducers, gall mites can form a wide variety of gall structures and induce galls on different parts of the host plant [2]. The physical stimuli of various types of gall mites are almost identical, with all of them using their stylets to stimulate cells, affecting the stimulated area or surrounding area to form galls [37,38]. The complexity of the gall structure induced by gall mites is intermediate between that of nematode-induced galls and that of insect-induced galls [8]. Moreover, Chetverikov [9] pointed out that an ideal model for studies of gall formation should satisfy the following criteria: (1) relatively simple cultivation in the laboratory; (2) a short life cycle and rapid gall formation; and (3) the presence of data on regulatory genes of the selected host plant in genome databases. Most galls are formed on woody plants [39], and the rearing of gall inducers in the laboratory is very difficult [36], so most samples, including galls and/or gall inducers used for research on the gall induction mechanism, are collected in the field, but not in the laboratory, and are much less involved in dynamic changes [28,33,40,41,42]. It is clear that, research on the gall formation mechanism has been difficult to conduct for a long time. Therefore, compared to galling insects with diverse feeding habits, life cycles and species, gall mite system interactions with plants should be a good system for studying the mechanism of gall formation.

In the current study, we examine a mite gall-inducing system to study the mechanisms underlying gall formation through experimental manipulations. Wolfberry, *Lycium barbarum* (Solanaceae) (Figure 1A), a traditional herb widely cultivated in Northwest China, is plagued by the gall mite *Aceria pallida* (Acari: Eriophyidae) (Figure 1B), which usually induces galls on young leaves and reproduces within them during the wolfberry growing season [43]. To date, we have succeeded in the long-term rearing of seedings of *L. barbarum* and *A. pallida* in the laboratory from a single female gall mite capable of inducing a gall suitable for herself and her progeny; the mites spend 2~3 d on average inducing a gall on wolfberry in the growth chamber [44,45]. To investigate the gall induction process, the histological and molecular changes during gall induction and development were investigated by inoculating the gall mite *A. pallida* onto wolfberry seedlings. Then, gall mite population dynamics, gall morphology and histological changes of wolfberry were observed. Meanwhile, the genetic and metabolic dynamics of wolfberry leaves were analyzed by transcriptomics and metabolomics, and the phytohormone (AUXs, CKs) concentrations of wolfberry at different developmental stages and gall mites were quantified by liquid chromatography–tandem mass spectrometry (LC–MS/MS).

## 2. Results

### 2.1. Alteration to L. barbarum Morphology and Population Dynamics of A. pallida

Most of the galls induced by *A. pallida* were on the leaves and showed a cystic shape, the color of which varied from yellowish green to green and then purple at different stages (Figure 2A–C). The average gall diameter varied from 0.74 mm at 3 d to 2.92 mm at 30 d, and the rapid gall growth period was days 3~9, with a growth rate of 86.08%, followed by 12.62% from 12 d to 18 d and finally by 5.84% from 21 d to 30 d (Figure 2D).

*A. pallida* can induce the gall by one or multiple mites, but always by a single one (Figure 2E), and then live and reproduce inside it (Figure 2F,G). The number of mites grew rapidly from 1.33 per gall to 353.33 per gall in 30 d. The increase was mainly concentrated within the first 18 d, with a compound growth rate (CGR) of 142.81% per 3 days but, after that, the CGR was only 33.65% per 3 d (Figure 2H). Moreover, the egg number peaked at 18 d, with an average of 128.87 per gall, and gradually decreased thereafter. The egg number at 18 d was 6.67- and 0.20-fold higher than that at 6 d and 30 d, respectively (Figure 2H).

There was a significant positive correlation between gall size and mite population size (r = 0.848, *p* < 0.01). According to the variation in gall size and mite population size, the galls used in the following study were divided into three developmental stages: (1) the induction stage, representing 3 d after mite infestation; (2) the growth stage, representing 9 d after mite infestation; and (3) the maturation stage, representing 18 d after mite infestation.

### 2.2. Histologic Structures of Galls Induced by A. pallida

Both the upper epidermal cell area of galls (G-EC) and the surrounding ungalled epidermal cell area (UG-EC) (Figure 3A) showed an increasing trend over time, although no significant differences were found between them at any developmental stages (Figure 3D). However, the upper epidermal cell length/width of galled tissue (G) was significantly greater than that of ungalled tissue (UG) beginning at 3 d, and the cell length/width of G-EC was 1.86-fold greater than that of UG-EC at 18 d (Figure 3E). With gall development, the palisade and spongy parenchyma within the gall were gradually replaced by nutritive tissue (NT) near the mite feeding site and storage tissue (ST) far from the feeding site (Figure 3B,C). The cells of NT were smaller than those of ST, their cytoplasm appeared to be denser than that of other cells within galls and their number showed a significant increase over time. The cell layers of galls (G-CL) increased significantly beginning at 9 d (Figure 3B), while the cell layers of the surrounding ungalled tissues (UG-CL) showed no obvious change; compared with those in UG-CL, the cell layers in G-CL were 1.72-fold greater at 9 d and 2.82-fold greater at 18 d (Figure 3F).

### 2.3. Alteration of L. barbarum Transcripts and Metabolites by A. pallida

In total, 97,302 transcript contigs were obtained with a total length of 6.57 G base pairs (bp) and an average length of 1003 bp. The GC content was 43.72%. After multiple filtering steps (see Section 4.4), 9246 unigenes were retained for final analysis. Principal component analysis (PCA) divided transcripts from the three infestation stages into two groups, the ungalled group and the galled group (Figure 4A), indicating that the transcript profiles of ungalled and galled tissues were significantly different. The number of differentially expressed genes (DEGs) between ungalled and galled tissues increased gradually with gall development, with 3527, 4675 and 4958 DEGs identified between ungalled and galled tissues at 3 d, 9 d and 18 d, respectively (Figure 4B). The number of upregulated DEGs was greater than that of downregulated DEGs at all developmental stages; the highest proportion of upregulated DEGs occurred at 9 d, and the ratio of upregulated/downregulated DEGs at 3 d, 9 d and 18 d was 1.68, 1.88 and 1.24, respectively (Figure 4B). These DEGs were mainly involved in photosynthesis, cellular respiration, substance transportation, phytohormone action, cell division and cell wall organization, carbohydrates, amino acids, fatty acids and secondary metabolism pathways (Table 1). Most of the DEGs involved in photosynthesis were downregulated, while most of the DEGs involved in cellular respiration and transportation, and the metabolism of materials such as carbohydrates, amino acids and fatty acids, were upregulated (Table 1).

A total of 1735 known compounds from 16 classes were detected in *L. barbarum* leaves by LC–MS/MS, of which 1111 compounds were detected under positive ion mode and 624 compounds were detected under negative ion mode. PCA divided metabolites from the three infestation stages into two groups, the ungalled group and the galled group, indicating that the metabolites from ungalled and galled tissues were significantly different; in addition, the difference between the ungalled group and galled group increased over time, indicating that the metabolic differences between ungalled and galled tissues widen with gall development (Figure 4C). The numbers of differential metabolites (DMs) at 3 d and 9 d were similar, but both were smaller than that at 18 d; 612, 596 and 940 DMs were identified between ungalled and galled tissues at 3 d, 9 d and 18 d, respectively (Figure 4D). The number of upregulated DMs was larger than that of downregulated DMs at all developmental stages; the highest proportion of upregulated DMs was observed at 18 d, and the ratio of upregulated/downregulated DMs was 1.10 at 3 d, 1.18 at 9 d and 1.76 at 18 d (Figure 4D). These DMs could mainly be classified into eight groups associated with carbohydrates, amino acids, fatty acids, lipids, alkaloids, organic acids, terpenes and phenolics (Table 2). The DMs from the carbohydrate, fatty acid, lipid, and organic acid groups were significantly upregulated at the three stages, the DMs from the amino acid, alkaloid and phenolic groups were significantly upregulated at 18 d, while the DMs from the terpene group were mainly downregulated (Table 2).

Combining the results for the transcriptome and metabolome, it could be seen that (1) the ungalled tissues and galled tissues differed distinctly and could be separated easily at both the gene and metabolite level and, (2) overall, the influence of the gall mite *A. pallida* on wolfberry strengthened with time, peaking at 18 d after infestation.

### 2.4. Manipulation of Hormone Balance and Cell Development in Galls by A. pallida

Cell division plays a significant role in gall development and the determination of gall shape [46]. A typical cell division cycle includes four phases: the G1 stage, S stage, G2 stage and M stage. In addition, cell wall formation is urgently needed for cytokinesis in plants [47], with the cell wall consisting mainly of polysaccharides such as cellulose, hemicellulose, pectin and several classes of proteins [30]. In the present study, the genes related to cell division were significantly upregulated in galled tissues over time (Figure 5A), especially genes involved in the S and M stages. Cyclin-dependent kinase (CDK), which regulates cyclin synthesis, was also markedly upregulated during gall development (Figure 5B). Furthermore, most of the genes involved in the components of the cell wall and its organization, such as lignin, cellulose, hemicellulose, pectin and cell wall protein synthesis, were distinctly upregulated in galled tissues at 3 d and 9 d (Figure 5B).

Cell hyperplasia (cell division) and hypertrophy (cell growth) are the two main processes of gall initiation and development, which points toward the implication of AUXs (specifically IAA) and CKs during gall formation [4]. A limited number of IAA and CKs metabolism genes were annotated in this study, based on the known pathways for them. Most of the genes that regulate IAA synthesis were significantly downregulated in galled tissues (Figure 5A) [48], including amidase 1 (AMI1), which was significantly downregulated at 9 d and 18 d, and aldehyde oxidase (AO) and flavin-dependent monooxygenase (YUC), which were significantly downregulated at all stages of development. Furthermore, the genes regulating IAA-amino acid conjugate synthase (GH3), which respond to a high concentration of IAA, were downregulated or showed no significance at 3 d and 9 d (Figure 5C). The genes involved in CKs synthesis were not annotated in galled tissues, but genes regulating CKs interconversion [49], such as adenosine kinase (AK) and zeatin-O-glucosyltransferase (ZOGT), and genes encoding cytokinin dehydrogenase (CKX), which is involved in CKs degradation at high concentrations of CKs, were significantly upregulated in galled tissues (Figure 5C). The pattern of GH3 and CKX indicated that IAA synthesis weakened and CKs degradation strengthened in galled tissues.

The AUX and CKs contents in ungalled tissues showed a decreasing trend over time, but most such contents in galled tissues were higher than those in ungalled tissues, and higher levels of AUXs and CKs were detected in the gall mite *A. pallida* than in plant tissues (Figure 6). Only one type of AUX, IAA, was detected in both ungalled and galled tissues. The IAA concentration in ungalled and galled tissues showed a decreasing trend with gall development, but its concentration in galled tissues was significantly higher than that in ungalled tissues at 3 d (1.42-fold) and 18 d (1.82-fold). Six types of CKs were detected in ungalled and galled tissues, including IP, IPR, *t*ZR, *c*Z, *c*ZR and BAPR. No significant difference in the IP, IPR or *c*Z content was found between ungalled and galled tissues at 3 d and 9 d, but their contents in galled tissues were significantly (2.48-fold and 34.46-fold, respectively) higher than those in ungalled tissues at 18 d. Interestingly, *c*Z was not detected in ungalled tissues at 18 d. The *c*ZR content in galled tissues was always higher than that in ungalled tissues, and significance was observed at 9 d and 18 d, being 1.61-fold and 36.11-fold higher than in ungalled tissues, respectively. The *t*ZR content in galled tissues was also higher than that in ungalled tissues at all developmental stages; it was 4.45-fold higher than that in ungalled tissues at 3 d and was not detected in ungalled tissues at 9 d and 18 d. In addition, BAPR in galled tissues was detected only at 9 d and 18 d, when it was 0.07 and 2.84 ng·g^−1^, respectively. Two types of AUXs (IAA, IBA) and ten types of CKs (IP, IPR, *c*Z, *c*ZR, *t*Z, *t*ZR, DHZ, BAP, BAPR, kinetin (K)) were detected in the gall mites. Most of their contents in mites were significantly higher than those in galled tissues except for BAPR, of which the IAA content was 1.69-fold, the IP content was 4826.56-fold, the IPR content was 28.99-fold and the *c*Z content was 592.93-fold higher than that in plant tissues.

### 2.5. Modification of Energy Metabolism in Galls by A. pallida

Chloroplasts and mitochondria are the two powerhouses of plant cells; the former is responsible for photosynthesis, and the latter is responsible for energy conversion [50]. In the present study, the genes related to photosynthesis were significantly downregulated in galled tissues at all developmental stages and showed an increasing inhibitory trend (Figure 7A), including genes regulating chlorophyll synthesis and photosystem II (PSII), photosystem I (PSI), the cytochrome b 6/f complex (Cytb 6/f), ATPase and the Calvin cycle (Figure 7B). However, genes regulating the tricarboxylic acid (TCA) cycle and oxidative phosphorylation were significantly upregulated in galled tissues during gall development (Figure 7A), with citrate synthase (CSY), aconitase (ACO), oxoglutarate dehydrogenase (ODH) and succinate dehydrogenase (SDH), as well as four oxidoreductase complexes (I, III, IV, and V), being significantly upregulated (Figure 7C). In addition, the genes that regulate transmembrane transportation were also significantly upregulated in galled tissues at all stages (Figure 7A), including the genes related to carbohydrate, protein and ion transport (Figure 7D). These results suggested that photosynthetic activity was suppressed, but cellular respiration and substance transport were enhanced, during gall development.

### 2.6. Changes in Nutrient Metabolism in Galls Induced by A. pallida

Gall inducers can manipulate the source–sink dynamics of plants, manifesting as galled tissues usually containing high concentrations of nutritive substances such as sugars, proteins, lipids and other nutrients [9]. In this study, the genes involved in carbohydrate synthesis were significantly upregulated in galled tissues at all stages (Figure 8A), including the genes regulating sucrose, glucose, mannose and starch synthesis and the genes involved in gluconeogenesis (Figure 8B). The genes involved in nonaromatic amino acid synthesis (glutamate and aspartate groups) and the genes encoding aminotransferases in galled tissues were markedly upregulated during gall development (Figure 8B). However, the genes involved in aromatic amino acid synthesis (shikimate group) in galled tissues were mainly downregulated (Figure 8B). For metabolism, the abundance of carbohydrates and amino acids, and their derivatives in galled tissues, also significantly increased at all developmental stages (Figure 8A), and the alteration peaked at 18 d (Figure 8C). Both the transcriptome and metabolome results showed that the nutrients in galled tissues increased with gall development.

### 2.7. Defensive Responses of L. barbarum to A. pallida

Host plants initiate various defense responses against gall inducers when infested, including direct defenses (the accumulation of secondary metabolites) and indirect defenses (the release of volatile components) [9]. In the present study, most of the genes involved in the PPP pathway and defense response were significantly upregulated in galled tissues (Figure 9A). The genes related to the PPP pathway were significantly upregulated in galled tissues at 9 d and 18 d, such as genes encoding phenylalanine ammonia lyase (PAL) at 18 d, cinnamate 4-hydroxylase (C4H) at 9 d and 18 d and 4-coumaroyl-CoA ligase (4CL) at 9 d (Figure 9B). In addition, the genes that regulate flavonoid synthesis were markedly upregulated in galled tissues at three stages, including flavanone 3-hydroxylase (F3H) and anthocyanin 3-O-glucosyltransferase (3GT) (Figure 9B). For metabolism, the abundance of flavonoids and coumarins significantly enhanced in galled tissues (Figure 9A), and the alteration peaked at 18 d (Figure 9C).

## 3. Discussion

This study systematically and comprehensively revealed the morphological dynamics of galls induced by the gall mite *A. pallida*, as well as the gall mite’s population dynamics and impacts on host transcripts, metabolites and phytohormones during gall formation. The gall induced by *A. pallida* is a closed cystic gall with a structure of storage tissues outside and nutritive tissues inside, developed by the cell elongation of epidermal cells and hyperplasia of mesophyll cells. The galls grew quickly within 9 d, and the mite population size increased rapidly within 18 d. Although most of the genes involved in phytohormone synthesis were distinctly downregulated, IAA and CKs were greatly enhanced in galled tissues, which may be mainly due to the mites’ contribution to phytohormones. The gall mite greatly influenced the chlorophyll synthesis and photosynthetic capacity of wolfberry but could enrich nutrients by transmembrane transportation, which was supported by the significant upregulation of most genes involved in the TCA cycle, oxidative phosphorylation process, transmembrane transportation and carbohydrate and amino acid synthesis, and the significant increase in the abundance of carbohydrates and amino acids, and their derivatives, in galled tissues. As a defense against the gall mite, wolfberry activated a defensive response and accumulated secondary metabolites, which were supported by the significant upregulation of most of the genes involved in the defensive response process and PPP pathway, and by a notable increase in the relative abundance of flavonoids and coumarins in galled tissues. Therefore, it could be seen that the gall mite *A. pallida* manipulated its host plant via morphology and metabolism, with the potential to contribute to phytohormones (IAA and CKs) during gall induction.

The formation and development of galls usually involve a series of cytological changes, such as cell hyperplasia, cell hypertrophy and cell redifferentiation [2]. In the current study, the upper epidermal cell length/width and cell layers of galls significantly increased, and genes related to the cell cycle were distinctly upregulated in galled tissues, indicating that the formation of galls induced by the gall mite *A. pallida* involved cell division. Similarly, cell hyperplasia was also observed in galls induced by the psyllid *Pseudophacopteron longicaudatum* [30]. During gall information, cell wall fortification is commonly inhibited, and cell expansion is promoted [24]. The genes involved in cell wall organization were strikingly upregulated in galled tissues; this was conducive to cell expansion in galled tissues induced by *A. pallida*. Cell wall modification was consistently detected in galls induced by the nematode *Meloidogyne incognita* [51]. Since the stylets of gall mites mouthparts can insert into the cell wall mechanically, cut, drill and produce salivary secretions [37], the degradation of the cell wall may be closely related to the salivary secretions of *A. pallida*.

The common traits of galls with negative effects on photosynthetic capacity have been observed in several galling models [41,52,53]. In the present study, the genes involved in chlorophyll synthesis and photosynthesis were notably downregulated in galled tissues at each infestation stage, suggesting that the gall mite *A. pallida* markedly inhibited photosynthesis efficiency (especially the process of photophosphorylation). Similarly, the strong suppression of the genes encoding enzymes involved in the biosynthesis of photosynthetic pigments (carotenoids and chlorophylls) was also detected in galls formed by *Daktulosphaira vitifoliae* [33]. Obviously, the photoassimilates of galled tissues are barely sufficient to support the galls, and the nutrition required by the gall mite *A. pallida* might thus be translocated directly or indirectly from the adjacent ungalled tissues around galled tissues. This hypothesis was supported by the genes involved in transmembrane transportation being significantly upregulated in galled tissues; thus, the gall mite *A. pallida* realized a shift from autotrophy to heterotrophy during gall induction. This phenomenon was in accordance with the characteristics of gall induction by the psyllid *Nothotrioza myrtoidis* and the gall midge *Bruggmanniella* sp., although there was less chlorophyll and more polysaccharide, or soluble sugar, accumulation in galled tissues [14,41]. A few genes involved in the Calvin cycle were found to be significantly upregulated in galled tissues compared with ungalled tissues. Considering that there were no stomas in the inner galled tissues, this promotion of the Calvin cycle may be attributable to the CO_2_ produced during gall mite respiration inside the gall, as shown in the galls of *D. vitifoliae* and *Schlechtendalia chinensis* [15,33].

AUXs and CKs are classes of phytohormones that induce cell growth and division, so the most general hypothesis is that gall formation is triggered by the action of phytohormones such as IAA and CKs [4]. Many recent works have shown that galled tissues are rich in IAA and CKs [28,40]. Exogenous CKs + IAA injections have been confirmed to lead to gall-like growth in *Capsicum annuum* [54]. In the current study, most of the genes involved in IAA synthesis were significantly downregulated in galled tissues, while genes regulating CKs conversion were significantly upregulated. In addition, the contents of IAA, *t*ZR and *c*ZR in galled tissues were dramatically higher than those in ungalled tissues at most infestation stages, indicating that higher contents of IAA and CKs existed in galled tissues during gall development. A similar pattern was also found in galls induced by *Eurosta solidagini* and *Cicadulina bipunctata* and leaves infested by the leaf-mining insect *Phyllonorycter blancardella* [11,27,55,56]. Active IAA and CKs have been identified in several gall inducers or their salivary secretions [35,57,58]. For example, Yamaguchi et al. [42] demonstrated that a galling insect was able to synthesize IAA from tryptophan. Thus, we reasoned that the enhanced content of IAA and CKs in galled tissues might have originated from the gall mite *A. pallida*. IAA can promote cell elongation and the maintenance of apical dominance by inhibiting the formation of lateral buds, and CKs can lead to cell division and promote the growth of plant tissues [59]; therefore, they might jointly contribute to the gall formation induced by *A. pallida*, in which IAA promotes epidermal cell elongation and CKs promote mesophyll cell division.

Gall inducers commonly change the chemical profile of their host plant tissues during gall induction and development [25]. In this study, the gall mite *A. pallida* needed a large amount of plant resources to develop and reproduce, and this demand for energy and carbon was reflected in the numerous genes involved in carbohydrate and amino acid synthesis being distinctly upregulated in the galled tissues at all developmental stages, which was also supported by the abundance of carbohydrates and amino acids and their derivatives being significantly enhanced in galled tissues compared with ungalled tissues. The same pattern was detected in flower galls induced by fig wasps, with dramatic upregulation of the transcripts associated with carbohydrate metabolism [17]. The genes involved in the PPP pathway were significantly upregulated in galled tissues at 9 d and 18 d, especially PAL, C4H and F3H enzymes, which play major roles in the biosynthesis of phenolic acids, such as flavonoids and coumarins [60], suggesting that the defense of wolfberry against *A. pallida* strengthened with infestation time, which was supported by the metabolism results: the abundance of flavonoids and coumarins prominently increased in galled tissues. Correspondingly, there were also much higher sugar, amino acid and phenol levels in the galls induced by the psyllid *Pauropsylla depressa* [61]. Some researchers have confirmed that tissues near the center of galls are usually rich in nutrient compounds, but that high levels of phenolics frequently accumulate outside of the gall [18,62], which could reduce the chemical defense of the host plants and facilitate the growth and development of gall inducers [25,31]. Phenolic acids might play roles in gall formation; for instance, flavonoids can act as transport regulators of IAA during gall formation [63], and coumarins (scopoletin) can act as IAA oxidase inhibitors [64]. Hence, we speculated that the elevated levels of phenolics in galled tissues of *A. pallida* not only play a defensive role but are also involved in the promotion of gall formation.

## 4. Materials and Methods

### 4.1. Plant and Mite Resources and Their Rearing Conditions

Seeds of *L. barbarum* were collected from fruits in Zhongning (37°29′ N, 105°42′ E), Ningxia Hui Autonomous Region, China, in 2020. The wolfberry seeds were sown in plastic trays (54 cm × 28 cm × 11 cm) containing potting soil media (Meihekou, Jilin Province, China) for germination and cultivated in growth chambers (25 ± 2 °C, 60 ± 5% relative humidity (RH), 14 L:10 D photoperiod, illumination: 18,000 l×). Approximately 6 weeks after sowing, every seedling with approximately 10 leaves was transplanted into a plastic cup (6.5 cm × 6.5 cm × 9 cm) and watered once every three days for continued cultivation under the abovementioned conditions. Approximately 2 weeks after transplanting, uniform seedlings with 15~20 leaves were selected for subsequent experiments.

The *A. pallida* used in this study were originally collected from *L. barbarum* in Zhongning (37°29′ N, 105°42′ E), Ningxia Hui Autonomous Region, China, reserved in the *L. barbarum* field at the Institute of Medicinal Plant Development (IMPLAD), Chinese Academy of Medical Sciences & Peking Union Medical College (CAMS & PUMC) (40°2′ N, 116°16′ E) in Beijing, China, and reared on wolfberry seedlings for more than 30 generations in growth chambers under the same conditions used for plant growth.

### 4.2. Preparation of Experimental Samples

A certain number of seedlings were prepared and divided into a control group and a treatment group. In the control group, no gall mites were inoculated. In the treatment group, the infestation of plants by gall mites was performed as follows: mature *A. pallida* galls (approximately 3 mm in diameter) cultured in the chamber were cut into 2 mm long and 0.05 mm wide slices (30~50 adult mites in each slice) under a stereomicroscope (Leica EZ4, Leica Microsystems, Wetzlar, Germany) [44], and then one slice was carefully attached to the tip of a seedling. Then, both the control and treatment groups were kept under the above rearing conditions.

The galls selected in this study were divided into 3 developmental stages according to the infestation time, gall size and mite population: (1) the induction stage, representing 3 d after mite infestation (Figure 10A); (2) the growth stage, representing 9 d after mite infestation (Figure 10B); and (3) the maturation stage, representing 18 d after mite infestation (Figure 10C). Ungalled leaves were divided into 5 equal parts, and the middle section was labeled D3C, D9C and D18C; the galls of leaves were dug out and labeled D3M, D9M and D18M.

Mites were collected from galls at 18 d after infestation. The galls were dissected under a stereomicroscope (Leica EZ4, Leica Microsystems, Wetzlar, Germany) and packed in a 50 mL centrifuge tube with sterile gauze, with approximately 15 g dissected galls per tube. To decrease the possibility of contamination and degradation, the tubes were injected with purified water, oscillated in an ice and water bath by ultrasound for 1 h, and subsequently centrifuged for 30 s at room temperature. Then, the upper water was absorbed by a micropipette (Eppendorf, Hamburg, Germany), and the lower sediment were the gall mites (Figure 10D).

### 4.3. Morphological and Anatomical Analysis

Galls of different developmental stages were photographed by stereomicroscopy (Leica EZ4, Leica Microsystems, Wetzlar, Germany). Ungalled and galled leaves in successive stages of development were fixed with FAA fixative solution (90 mL 50% ethanol, 5 mL glacial acetic acid, 5 mL formalin). All leaves were dehydrated with an ethanol series (50% ethanol for 1 h → 60% ethanol for 1 h → 70% ethanol for 1 h → 80% ethanol for 1 h → 90% ethanol for 1 h → 100% ethanol for 30 min → 100% ethanol for 30 min), hyalinized with xylene and then embedded in paraffin (MPS/P2, SLEE, Mainz, Germany). Transverse serial sections, 10~12 μm thick, were cut with a rotary microtome (CUT 6062, SLEE, Germany). Slices were dyed with safranine and Fast Green for 6~20 s, dehydrated and made into permanent slides. The yielded slices were observed and photographed by a polarizing microscope with Tissue FAXS Plus software (Axio Imager Z2, Zeiss, Oberkochen, Germany).

The cell area and cell layers were calculated and counted by Digimizer 6.0 software. There were 3 replicate slices for each treatment at each time point (3, 9 and 18 d after mite infestation). To reduce the differences between individual leaves, the ungalled tissues surrounding galled tissues were taken as the control.

### 4.4. RNA Extraction, Transcriptome Sequencing and Analysis

Total RNA was isolated from frozen samples of plant tissues (3 replicates, 100 mg each) and gall mites (3 replicates, 30 mg each) using a TRIzol extraction kit (Tiangen Biotechnology, Beijing, China) following the manufacturer’s instructions. The integrity and purity of RNA samples were assessed using an Agilent Bioanalyzer 2100 system (Agilent Technologies, Santa Clara, CA, USA) and a NanoPhotometer spectrophotometer (Thermo Fisher Scientific, Waltham, MA, USA), respectively. RNA samples of acceptable quality were used to construct non-strand-specific sequencing libraries with the TruSeq RNA Sample Prep Kit (Illumina, San Diego, CA, USA). These libraries were sequenced using an Illumina NovaSeq 6000 by Novogene Co., Ltd., in Beijing, China (https://cn.novogene.com/ (accessed on 25 September 2022)).

Raw RNA-Seq reads were processed to remove adapters, reads containing more than two N bases and low-quality sequences using Trimmomatic [65]. De novo assembly of the clean sequence reads was carried out by Trinity v2.4.0 [66]. The transcript obtained by Trinity was taken as the reference sequence (Ref), the clean reads of each sample were mapped to Ref using RESM v1.2.15 [67], and then raw read counts for each transcript were derived. The read counts were imported into the R package DESeq2 v1.6.3 [68] for fragments per kilobase exon per million mapped fragments (FPKM) normalization. To exclude possible mite tissue contaminants from the galled tissue transcript files, a blast search of the protein sequences of galls and mites was carried out; subsequently, any matching sequences in the gall transcript file were removed. Finally, any transcript contigs with three replicates averaging <10 FPKM across all groups were excluded. Unigenes were annotated against NCBI nonredundant protein sequences (Nr) for function and against the Gene Ontology (GO) and Mercator4 protein function mapping databases for pathways [69].

DEGs were identified by the following criteria: absolute FPKM value more than 10, fold change (FC) between the treated group and control group greater than 2 (log2 ratio > 1 or < −1) and *p* value less than 0.05.

### 4.5. Metabolic Analysis by Liquid Chromatography Coupled to Tandem Mass Spectrometry (LC–MS/MS)

An amount of 100 mg plant tissue powder was individually ground with liquid nitrogen, and the homogenate was resuspended with prechilled 80% methanol (MeOH) by well vortex. The samples were incubated on ice for 5 min and then centrifuged at 15,000 r·s^−1^ and 4 °C for 20 min. Some of the supernatant was diluted to a final concentration containing 53% MeOH by LC–MS grade water. The samples were subsequently transferred to a fresh tube and then centrifuged at 15,000 r·s^−1^ and 4 °C for 20 min. Finally, the supernatant was injected into the LC–MS/MS system for analysis.

UPLC–MS/MS analyses were performed using a Vanquish UHPLC system coupled with an Orbitrap Q Exactive^TM^ HF mass spectrometer (Thermo Fisher Scientific, Bremen, Germany). Samples were injected into a Hypesil Gold column (100 mm × 2.1 mm, 1.9 μm, Thermo Fisher Scientific, Bremen, Germany) using a 12 min linear gradient at a flow rate of 0.2 mL·min^−1^. The eluents for the positive polarity mode were eluent A (0.1% formic acid in water) and eluent B (MeOH). The eluents for the negative polarity mode were eluent A (5 mM ammonium acetate, pH 9.0) and eluent B (MeOH). The solvent gradient was as follows: 0~1.5 min, from 2% to 2% B; 1.5~3 min, from 2% to 85% B; 3~10 min, from 85% to 100% B; 10~10.1 min, from 100% to 2% B; 10.1~12 min, from 2% to 2%. The mass spectrometer was operated in positive/negative polarity mode with a spray voltage of 3.5 kV, a capillary temperature of 320 °C, a sheath gas flow rate of 35 psi, an auxiliary gas flow rate of 10 L·min^−1^, an S-lens RF level of 60, and an auxiliary gas heater temperature of 350 °C.

The raw data files generated by UPLC–MS/MS were processed using Compound Discoverer 3.1 (CD3.1, Thermo Fisher Scientific, Bremen, Germany) to perform peak alignment, peak selection, and quantitation for each metabolite. The main parameters were set as follows: retention time tolerance, 0.2 min; actual mass tolerance, 5 ppm; signal intensity tolerance, 30%; signal/noise ratio, 3; and signal intensity, minimum. After that, peak intensities were normalized to the total spectral intensity. The normalized data were used to predict the molecular formula based on additive ions, molecular ion peaks and fragment ions. Then, peaks were matched with the mzCloud (https://www.mzcloud.org/ (accessed on 20 September 2022)), mzVault and MassList databases to obtain accurate qualitative and relative quantitative results. Statistical analyses were performed using the statistical software R (version 3.4.3), Python (version 2.7.6) and CentOS (version 6.6). When data were not normally distributed, normal transformations were attempted using the area normalization method.

DMs were identified by the following criteria: FC between the treated group and control group greater than 2 (log2 ratio > 1 or < −1) and *p* value less than 0.05.

### 4.6. Phytohormone Quantification by LC–MS/MS

The plant tissue and mite samples were ground into powder by a mixer mill (MM 400, Retsch, Haan, Germany) at 30 Hz for 1 min. Then, 50 mg powder was weighed into a 2 mL plastic microtube, frozen in liquid nitrogen and dissolved in 1 mL MeOH/water/formic acid (15:4:1, *v*/*v*/*v*). An amount of 10 μL internal standard mixed solution (100 ng·mL^−1^) was added to the extract as an internal standard (IS) for quantification. The mixture was vortexed for 10 min and then centrifuged for 5 min (12,000 r·min^−1^, 4 °C). The supernatant was transferred into clean plastic microtubes, dried by evaporation under a flow of nitrogen gas, dissolved in 100 μL 80% MeOH (*v*/*v*) and filtered (PTFE, 0.22 μm, Anpel, Shanghai, China) for further LC–MS/MS analysis.

HPLC-grade acetonitrile (ACN) and MeOH were purchased from Merck (Darmstadt, Germany). Milli-Q water (Millipore Corp., Bradford, MA, USA) was used in all experiments. All of the standards were purchased from Olchemim Ltd. (Olomouc, Czech Republic) and IsoReag (Shanghai, China). Acetic acid and formic acid were purchased from Sigma–Aldrich (St. Louis, MO, USA). Stock solutions of standards were prepared at a concentration of 1 mg·mL^−1^ in MeOH. All stock solutions were stored at −20 °C. The stock solutions were diluted with MeOH to working solutions (0.01 ng·mL^−1^, 0.05 ng·mL^−1^, 0.1 ng·mL^−1^, 0.5 ng·mL^−1^, 1 ng·mL^−1^, 5 ng·mL^−1^, 10 ng·mL^−1^, 50 ng·mL^−1^, 100 ng·mL^−1^, 200 ng·mL^−1^, 500 ng·mL^−1^) before analysis.

The sample extracts were injected into an UPLC–ESI–MS/MS system (UPLC, ExionLC™ AD; MS, Applied Biosystems 6500 Triple Quadrupole, Applied Biosystems Inc., Foster, CA, USA) using a Waters ACQUITY UPLC HSS T3 C18 column (100 mm × 2.1 mm, 1.8 µm, Waters Corp., Milford, MA, USA). Water with 0.04% acetic acid (solvent A) and ACN with 0.04% acetic acid (solvent B) were used as gradients. The analytical conditions were as follows: 0~1 min, gradient from 5% to 5% B; 1~8 min, gradient from 5% to 95% B; 8~9 min, gradient from 95% to 95% B; 9.1~12 min, gradient from 95% to 5% B. The other settings were flow rate: 0.35 mL·min^−1^; temperature: 40 °C; and injection volume: 2 μL [70].

Linear ion trap (LIT) and triple quadrupole (QQQ) scans were acquired on a triple quadrupole–linear ion trap mass spectrometer (QTRAP), QTRAP^®^ 6500+ LC–MS/MS System, equipped with an ESI Turbo Ion-Spray interface, operating in both positive and negative ion modes and controlled by Analyst 1.6.3 software (AB Sciex, Waltham, MA, USA). The ESI source operation parameters were as follows: ion source, ESI+/−; source temperature, 550 °C; ion spray voltage (IS), 5500 V (positive), −4500 V (negative); and curtain gas (CUR), 35 psi [71]. Phytohormones were analyzed using scheduled multiple reaction monitoring (MRM). Data acquisitions were performed using Analyst 1.6.3 software (AB Sciex, Waltham, MA, USA). Multiquant 3.0.3 software (AB Sciex, Waltham, MA, USA) was used to quantify all metabolites. Mass spectrometer parameters, including the decluttering potentials (DPs) and collision energies (CEs) for individual MRM transitions, were determined with further DP and CE optimization. A specific set of MRM transitions was monitored for each period according to the metabolites eluted within this period.

### 4.7. Statistical Analysis

SPSS 21.0 software for Windows (IBM SPSS, Somers, NY, USA) was used to perform one-way analysis of variance (ANOVA) following Student’s *t* test (*p* < 0.05) and Pearson’s correlation analysis (*p* < 0.01). The bar charts were generated using GraphPad Prism 8.0 (http://www.graphpad.com/ (accessed on 20 March 2023)). PCA of transcripts and metabolites was performed via the Metware Cloud, a free online platform for data analysis (https://cloud.metware.cn (accessed on 20 March 2023)). Heatmaps were generated with TBtools software (https://github.com/CJ-Chen/TBtools/releases (accessed on 10 March 2023)). Transcripts and metabolites were placed into the metabolic pathways using the MapMan4 visualization toolkit (https://mapman.gabipd.org (accessed on 10 September 2022)).

## 5. Conclusions

This paper provides a first glimpse into the dynamic responses of host plants to gall mites during gall induction and development at the transcriptomic, metabolic and phytohormonal levels simultaneously, based on laboratory samples. The current results indicate that: (1) the gall mite *A. pallida* induces galls via cell elongation of the epidermis and the division of mesophyll cells; (2) it transforms the photo assimilative capacity of *L. barbarum* leaves into a nutrient sink by transmembrane transport and by enhancing nutrient synthesis; and (3) it does not significantly alter the expression of the genes involved in IAA and CKs synthesis in *L. barbarum*, but has the potential ability to synthesize these compounds. The significant economic importance of gall mites is associated with their ability to transmit phytopathogens and induce galls on plants, but their minute size, and the problems related to protracted maintenance of these mites in culture, make them rather inconvenient objects for examination [37]. We argue that *A. pallida* + *L. barbarum* is an ideal model for studying the gall induction mechanism of gall mites due to the convenience of long-term rearing in the laboratory. In conclusion, this study will pave the way for future studies of gall induction by mites and plant biotic interactions.

## Figures and Tables

**Figure 1 ijms-24-09839-f001:**
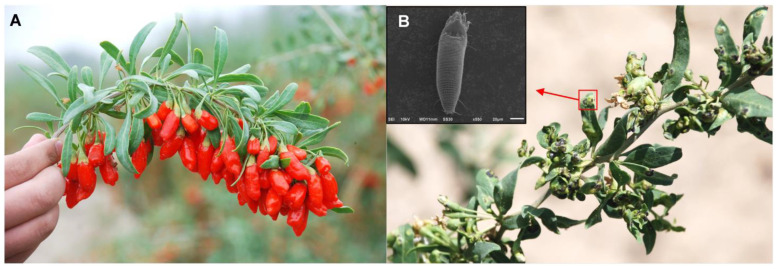
*Lycium barbarum* and the symptoms of leaves infested by *Aceria pallida*. (**A**) *L. barbarum*. (**B**) Galls induced by *A. pallida* on leaves of wolfberry, scale bar = 20 μm.

**Figure 2 ijms-24-09839-f002:**
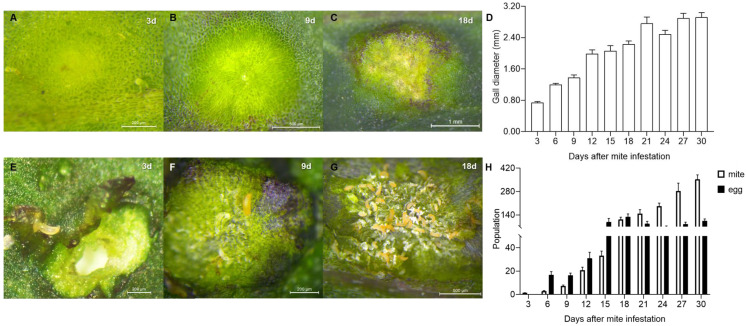
Growth dynamics of galls and the mite population. (**A**–**C**): Gall morphology at 3 d, 9 d and 18 d after infestation. (**D**): Growth dynamics of galls induced by *A. pallida* from 3 d to 30 d. (**E**–**G**): Mite population inside galls at 3 d, 9 d and 18 d after infestation. (**H**): Population dynamics of mites and their eggs per gall from 3 d to 30 d.

**Figure 3 ijms-24-09839-f003:**
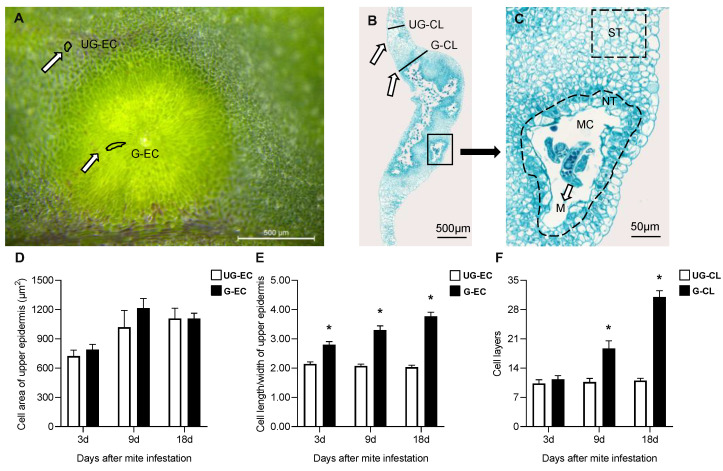
Cell area and morphology of galls induced by *A. pallida*. (**A**): Cell morphology of the upper epidermis of a galled leaf. Abbreviations: G-EC: epidermal cell of galls, UG-EC: epidermal cell of ungalled tissues surrounding the gall. Scale bar = 500 μm. (**B**): Vertical section of a galled leaf. Abbreviations: G-CL: cell layers of galls, UG-CL: cell layers of ungalled tissues surrounding the gall. Scale bar = 500 μm. (**C**): Cell types inside galls. Abbreviations: M: mite; MC: mite chamber; ST: storage tissue; NT: nutritive tissue. Scale bar = 50 μm. (**D**): Cell area of the upper epidermis of a galled leaf. The white column represents epidermal cells of ungalled tissues of the surrounding gall (UG-EC), and the black column represents epidermal cells of the gall (G-EC). (**E**): Cell length/width of the upper epidermis of a galled leaf. (**F**): Cell layers of a galled leaf. The white column represents the cell layers of galls (G-CL), and the black column represents the cell layers of ungalled tissues surrounding galls (G-CL). * indicates a significant difference between ungalled (UG) and galled (G) tissues at the same stage at the level of *p* < 0.05.

**Figure 4 ijms-24-09839-f004:**
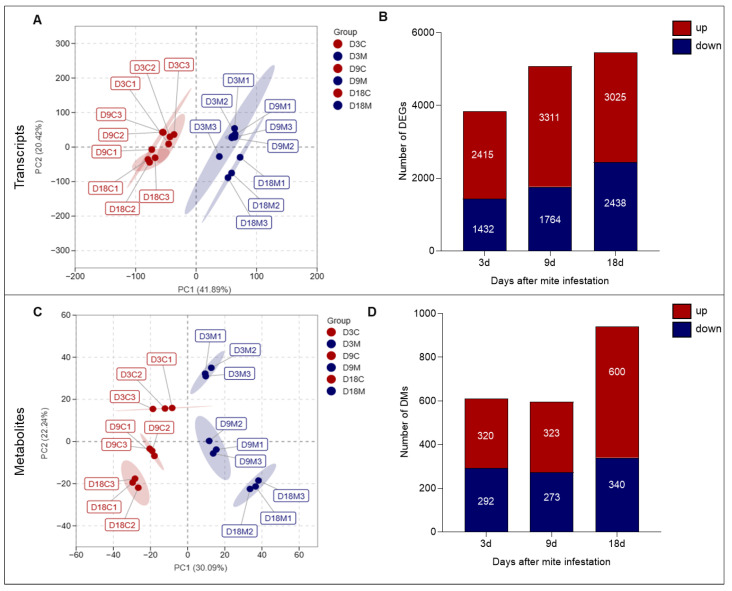
Overview of transcripts and metabolites in galled and ungalled tissues. (**A**) Principal component analysis (PCA) of transcripts. (**B**) Number of differentially expressed genes (DEGs). (**C**) PCA of metabolites. (**D**) Number of differential metabolites (DMs). In the PCA chart, red indicates ungalled tissues at 3 d (D3C), 9 d (D9C) and 18 d (D18C), and blue indicates galled tissues at 3 d (D3M), 9 d (D9M) and 18 d (D18M). In the column charts, red indicates the number of upregulated genes or metabolites, and blue indicates the number of downregulated genes or metabolites. The fold change threshold for the transcriptome and metabolome is log 2 > 1 or log 2 < −1.

**Figure 5 ijms-24-09839-f005:**
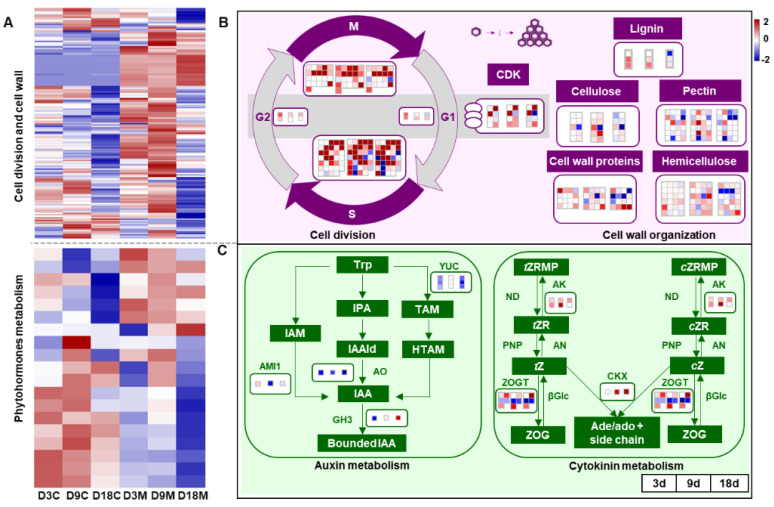
Transcriptional changes during cell development and phytohormone metabolism of galled tissues vs. ungalled tissues. (**A**) Heatmap of DEGs that regulate the cell division cycle, cell wall organization and phytohormone metabolism in ungalled tissues and galled tissues at 3 d, 9 d and 18 d after mite infestation. (**B**) DEGs involved in the cell division cycle and cell wall organization in galled tissues vs. ungalled tissues at 3 d, 9 d and 18 d after mite infestation. S: DNA replication of the cell cycle, M: mitosis of the cell cycle, G1: gap between mitosis and DNA replication, G2: gap between DNA replication and mitosis, CDK: cyclin-dependent kinase. (**C**) DEGs involved in auxin and cytokinin metabolism in galled tissues vs. ungalled tissues at 3 d, 9 d and 18 d after mite infestation. Abbreviations: Trp: tryptophan, IAM: indole-3-acetamide, IPA: indole-3-pyruvic acid, IAAld: indole-3-acetaldehyde, TAM: tryptamine, HTAM: N-hydroxy-tryptamine, IAA: indole-3-acetic acid, AMI1: amidase 1, AO: aldehyde oxidase, YUC: flavin-dependent monooxygenase, GH3: IAA-amino acid conjugate synthase, *t*ZRMP: *trans* zeatin riboside 5′-monophosphate, *t*ZR: *trans* zeatin riboside, *t*Z: *trans* zeatin, ZOG: zeatin-O-glucoside, *c*ZRMP: *cis* zeatin riboside 5′-monophosphate, *c*ZR: *cis* zeatin riboside, cZ: *cis* zeatin, AK: adenosine kinase, ND: 5′-nucleotidase, AN: adenosine nucleosidase, PNP: purine nucleoside phosphorylase, ZOGT: zeatin-O-glucosyltransferase, βGlc: β-glucosidase, CKX: cytokinin dehydrogenase. Each block in the map represents a single gene, red boxes represent upregulated genes and blue boxes represent downregulated genes. In the cytokinin metabolism pathways, genes regulating ND, PNP, AN and βGlc were not annotated in this study. The fold change thresholds of transcripts and metabolites were log 2 > 2 or log 2 < −2. The same below.

**Figure 6 ijms-24-09839-f006:**
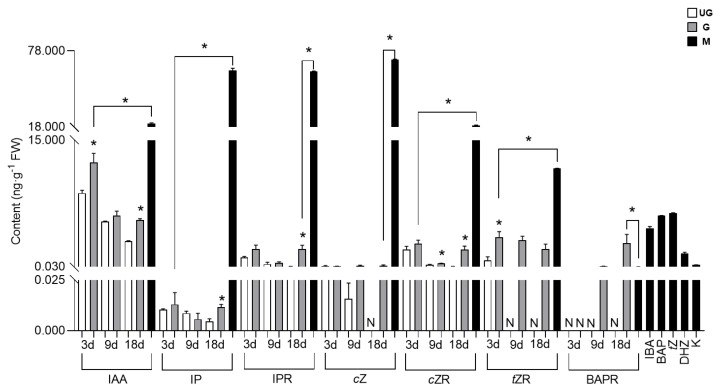
Quantified auxins and cytokinins contents of plant tissues and gall mites (ng·g^−1^). The white bar represents ungalled tissues (UG), the gray bar represents galled tissues (G) and the black bar represents gall mites (M). Values represent means ± standard errors (*n* = 3). Abbreviations: IAA: indoleacetic-3-acid; IP: isopentenyladenine; IPR: isopentenyladenosine; *c*Z: *cis* zeatin; *c*ZR: *cis* zeatin riboside; *t*ZR: *trans* zeatin riboside; BAPR: 6-benzylaminopurine riboside; IBA: indole-3-butyric acid; BAP: 6-benzylaminopurine; *t*Z: *trans* zeatin; DHZ: 6-benzylaminopurine riboside; K: kinetin. * indicates a significant difference between ungalled and galled tissues at the same stage or between gall mites and the plant tissue with the highest phytohormone content at the level of *p* < 0.05. N indicates not detected.

**Figure 7 ijms-24-09839-f007:**
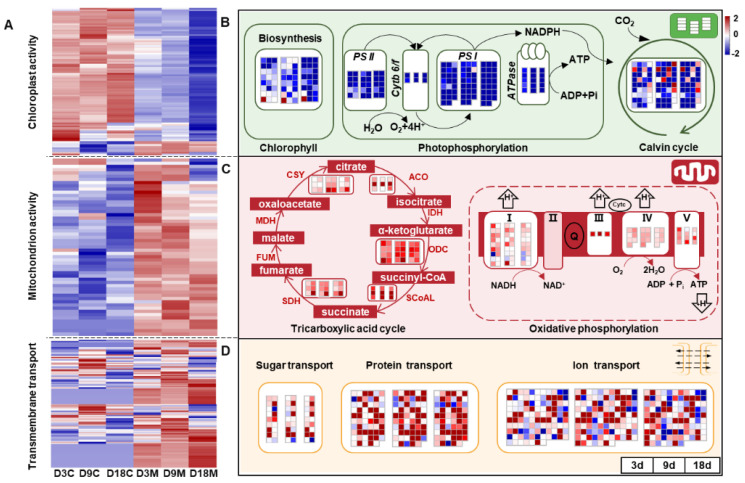
Transcriptional changes for energy metabolism and transmembrane transportation in galled tissues vs. ungalled tissues. (**A**) Heatmap of DEGs that regulate chloroplast activity, mitochondrion activity and transmembrane transportation in ungalled tissues and galled tissues at 3 d, 9 d and 18 d after mite infestation. (**B**) Energy metabolism in chloroplasts of galled tissues vs. ungalled tissues at 3 d, 9 d and 18 d after mite infestation. (**C**) Energy metabolism in mitochondria of galled tissues vs. ungalled tissues at 3 d, 9 d and 18 d after mite infestation. (**D**) Transmembrane transportation among cells of galled tissues vs. ungalled tissues at 3 d, 9 d and 18 d after mite infestation. Abbreviations: PSII: photosystem II, PSI: photosystem I, Cytb 6/f: cytochrome b 6/f complex, CSY: citrate synthase, ACO: aconitase, IDH: isocitrate dehydrogenase, ODH: oxoglutarate dehydrogenase, SCoAL: succinyl-CoA ligase, SDH: succinate dehydrogenase, FUM: fumarase, MDH: malate dehydrogenase. Genes regulating IDH, FUM and MDH in the TCA cycle pathways and oxidoreductase complexes II in the oxidative phosphorylation pathways were not annotated in this study.

**Figure 8 ijms-24-09839-f008:**
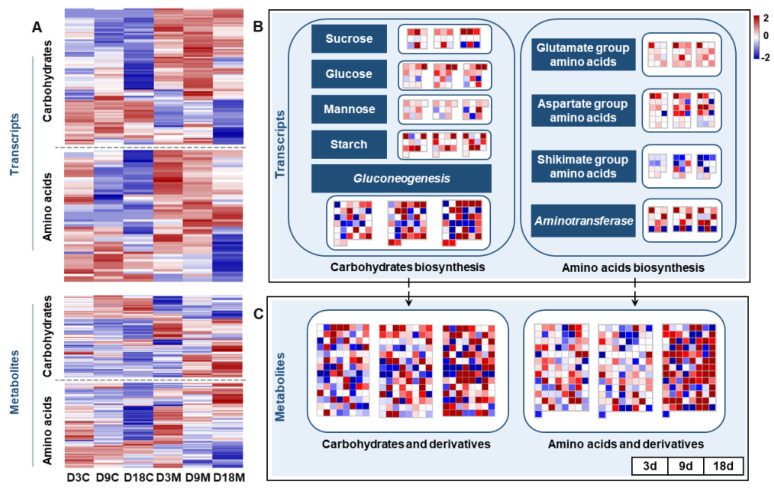
Transcriptional and metabolic changes in nutrient metabolism in galled tissues vs. ungalled tissues. (**A**) Heatmap of DEGs that regulate carbohydrate and amino acid synthesis, and DMs of carbohydrates and amino acids and their derivatives, in ungalled tissues and galled tissues at 3 d, 9 d and 18 d after mite infestation. (**B**) DEGs that regulate carbohydrate and amino acid synthesis in galled tissues vs. ungalled tissues at 3 d, 9 d and 18 d after mite infestation. (**C**) DMs of carbohydrates and amino acids and their derivatives in galled tissues vs. ungalled tissues at 3 d, 9 d and 18 d after mite infestation.

**Figure 9 ijms-24-09839-f009:**
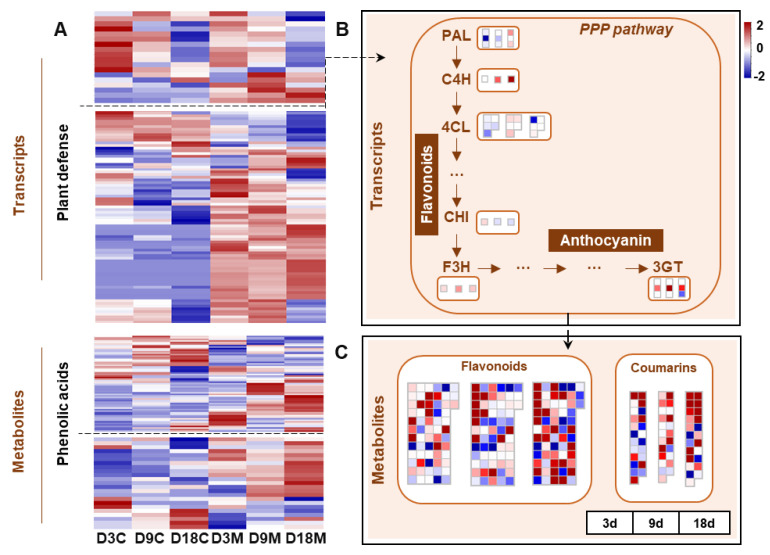
Transcriptional and metabolic changes for defensive metabolism in galled tissues vs. ungalled tissues. (**A**) Heatmap of DEGs involved in the defensive response process and DMs of flavonoids and coumarins in ungalled tissues and galled tissues at 3 d, 9 d and 18 d after mite infestation. (**B**) Transcriptional changes in the phenylpropanoid (PPP) pathway in galled tissues vs. ungalled tissues at 3 d, 9 d and 18 d after mite infestation. (**C**) DMs of flavonoids and coumarins in galled tissues vs. ungalled tissues at 3 d, 9 d and 18 d after mite infestation. Abbreviations: PAL: phenylalanine ammonia lyase, C4H: cinnamate 4-hydroxylase, 4CL: 4-coumaroyl-CoA ligase, CHI: chalcone isomerase, F3H: flavanone 3-hydroxylase, 3GT: anthocyanin 3-O-glucosyltransferase.

**Figure 10 ijms-24-09839-f010:**
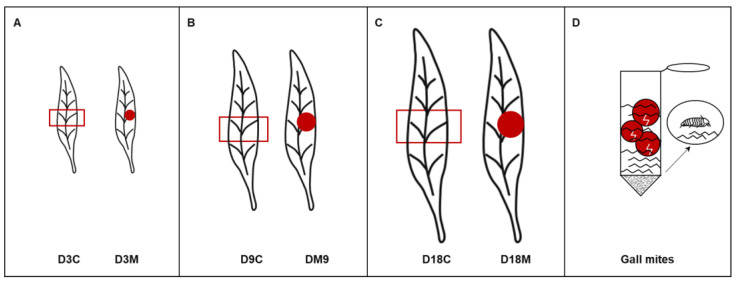
Sample acquisition for further analysis. (**A**) Ungalled leaf and galled leaf at 3 d after mite infestation. (**B**) Ungalled leaf and galled leaf at 9 d after mite infestation. (**C**) Ungalled leaf and galled leaf at 18 d after mite infestation. (**D**) Gall mite collection. The red box indicates the control group which is originated from middle section of an ungalled leaf, the red point indicates the treated group which is the gall induced by *A. pallida*.

**Table 1 ijms-24-09839-t001:** The differentially expressed genes (DEGs) involved in the main pathways of galled vs. ungalled tissues after infestation for 3 d, 9 d and 18 d (Based on Mercator4).

Pathway	Total DEGs			Upregulated			Downregulated		
	3 d	9 d	18 d	3 d	9 d	18 d	3 d	9 d	18 d
Photosynthesis	152	159	185	7	12	13	145	147	172
Cellular respiration	69	70	65	68	68	61	1	2	4
Vesicle trafficking	35	63	61	35	63	58	0	0	3
Protein translocation	31	46	39	29	38	25	2	8	14
Phytohormone action	23	44	53	18	25	28	5	19	25
Cell division	50	72	77	46	62	57	4	10	20
Cytoskeleton organization	25	35	26	24	33	21	1	2	5
Cell wall organization	14	25	26	9	16	9	5	9	17
Carbohydrate metabolism	28	45	45	21	36	32	7	9	13
Amino acid metabolism	24	41	44	14	27	25	10	14	19
Protein biosynthesis	184	277	199	180	222	139	4	55	60
Lipid metabolism	54	77	95	39	53	60	15	24	35
Secondary metabolism	19	21	29	11	13	16	8	8	13

**Table 2 ijms-24-09839-t002:** Main types of differential metabolites (DMs) in galled vs. ungalled tissues after infestation for 3 d, 9 d and 18 d, respectively.

Type	Total DMs			Upregulated			Downregulated		
	3 d	9 d	18 d	3 d	9 d	18 d	3 d	9 d	18 d
Carbohydrates	63	58	83	32	36	56	31	22	27
Amino acids	52	41	86	30	19	60	22	22	26
Fatty acids	36	42	53	25	27	34	11	15	19
Lipids	78	87	111	47	69	82	31	18	29
Alkaloids	117	101	169	46	43	105	71	58	64
Organic acids	47	54	76	26	31	54	21	23	22
Terpenes	58	52	77	29	18	38	29	34	39
Phenolics	60	61	92	31	31	56	29	30	36

## Data Availability

The raw data of transcripts that support the findings of this study are openly available in the NCBI Sequence Read Archive (SRA) under project PRJNA961943.

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
