# Peer review of "Integrated Transcriptome and Metabolome Dynamic Analysis of Galls Induced by the Gall Mite Aceria pallida on Lycium barbarum Reveals the Molecular Mechanism Underlying Gall Formation and Development"

_ijms, 2023, doi:10.3390/ijms24129839_

Round 1

Reviewer 1 Report

This is an excellent paper illuminating several aspects of the development of plant galls elicited by mites. All arthropod galls arise from the galler's ability to alter the expression of plant genes to modify morphological development, chemistry, physiology and hormone production/function. This paper is exceptional in providing significant insights on all of those impacts. Most papers limit their view to one or two. Many of the present results are similar to those for other studies of other galls formed by a variety of insects and arthropods. But this paper converges on those papers in multiple ways. It makes a major contribution to delimiting the suite of functions required for gall formation. We need every study of this type we can get if we are going to understand gall formation. I am mostly happy with the methods as described and have only a few questions below, mainly about their interpretations of results and some unclear wording. 

            First, some clarification issues:

            120 " A. pallida always induced the gall by a single one"  does this mean that each gall is formed by one mite? Wording is unclear

            271  "No significant difference in the IP, IPR or cZ content was found between ungalled and galled tissues at 3 d and 9 d, but their contents in galled tissues were significantly (2.48-fold and 34.46-fold, respectively) higher than those in ungalled tissues." Is something missing from this sentence? How can CK contents of galled tissue be significantly higher than ungalled tissue and yet there are no significant differences between gall and ungalled tissues?

            The IPR label missing from figure 6.

            In Fig 3 is the arrow from 4B to 4C meant to indicate that 4C is an enlargement of the area boxed in 4B? They don't look the same to me. If they are not the very same area, delete the arrow.

            In Fig 4 which clusters are galls and which are ungalled tissues? Can the PCs be characterized in terms of which genes are involved in each?

            About methods and interpretations: 

            542  The authors used Trinity to develop a reference transcritome: " De novo assembly of the clean sequence reads was carried out by Trinity v2.4.0. The transcript obtained by Trinity was taken as the reference sequence." Why didn't they use the Lycium genome?  Cao, YL., Li, Yl., Fan, YF. et al. Wolfberry genomes and the evolution of Lycium (Solanaceae). Commun Biol 4, 671 (2021). https://doi.org/10.1038/s42003-021-02152-8.

            A problem working with a non-model plant is a lack of functional annotation. Other authors have aligned the gene and/or protein sequences from their non-model plant with those of a better-annotated species, like Arabidopsis. While there is a risk of incorrect functional attributions, the risk may be worth it if more proteins can be characterized. I notice that quite a few genes/proteins in the authors' pathway analyses (e.g. Fig. 5) are missing. For example, up- or down-regulation is reported for only a very few genes in the hormone pathways. Were the missing ones not found, or were they not differentially expressed? If they weren't found using a well-annotated reference genome might improve this. In any case this question needs to be stated in the figure legend (i.e. missing or not DEG?). 

            Whether CK or AUX were being synthesized depends on whether limiting steps in their synthesis pathways were blocked. The authors claim that the genes whose expression differed between gall and ungalled tissue regulate these syntheses. But they provide no citation or other evidence that these (few) genes are a limiting step in the process. For example, is increased activity of AK alone really sufficient to increase CK concentrations? 

            The authors concluded (line 243) that their genetic evidence means that IAA synthesis weakened and CK degradation strengthened in galled tissues. I agree, but the reasoning should be that GH3 responds to high auxin concentrations and CKX responds to high CK concentrations. The activation of genes whose proteins conjugate those hormoned in response to their presence/concentration tells us that those hormones were present and abundant. That could be pointed out more directly. The paper is exceptional in obtaining this result AND chemical measures corroborating this claim. 

            Finding that particular functional groups (eg., GOs) are enriched in galls vs ungalled tissues is meaningless without an analysis of WHICH genes in those groups were up/down regulated and what each one does. This is because an upregulated gene can be a suppressor and a downregulated gene can be a promoter. For example, if we want to know whether differential gene expression altered photosynthesis in galls, the role of each gene when up- or down-regulated must be known, not just their numbers. For example, the fact that more genes involved in cell division were upregulated than downregulated does not mean that cell division was increasing. If the authors can support their claim about the DEGs in particular pathways (see above) that would help (and would require references). But for the functions they studied in less detail (Table 1) the numbers of up vs downregulated genes in that table really tell us nothing. They could identify key DEGs whose function is known to stretngthen conclusions from those data. That might increase the length of the paper a lot, though. They at least need to speculate on the meaning of any GO group's enrichment very carefully. 

Reviewer 2 Report

In this study, the morphology of gall was observed and gall formation was analyzed at the gene level and metabolite level by an integrated method using RNA seq and LC MS analysis.

The test method of this study is not original, but the formation mechanism of gall was analyzed by integrating each analysis.

The presented conclusion is thought to be consistent with the purpose, and the research method suitable for the purpose is thought to have been established. Tables and figures are also judged to be appropriately used.

This is a very interesting study. The experiment to prove the Gall's phenotype was judged to be very appropriate, and the considerations based on the results were also thought to be very well written.

Author Response

Thanks for your comments warmly.